# Macrophage Stimulated by Low Ambient Temperature Hasten Tumor Growth via Glutamine Production

**DOI:** 10.3390/biomedicines8100381

**Published:** 2020-09-26

**Authors:** Eun-Ji Lee, Tae-Wook Chung, Keuk-Jun Kim, Boram Bae, Bo-Sung Kim, Suhkmann Kim, Dongryeol Ryu, Sung-Jin Bae, Ki-Tae Ha

**Affiliations:** 1Department of Korean Medical Science, School of Korean Medicine, Pusan National University, Yangsan, Gyeongnam 50612, Korea; lej@pusan.ac.kr (E.-J.L.); kkc0704@pusan.ac.kr (B.-S.K.); 2Korean Medical Research Center for Healthy Aging, Pusan National University, Yangsan, Gyeongnam 50612, Korea; twchung@pusan.ac.kr (T.-W.C.); corona1814@pusan.ac.kr (B.B.); 3Korean Medicine (KM) Application Center, Korea Institute of Oriental Medicine (KIOM), 70 Chumdan-ro, Dong-Gu, Daegu 41062, Korea; 4Department of Clinical Pathology, DaeKyeung University, Gyeongsan 38547, Korea; biomed@tk.ac.kr; 5Department of Chemistry, Center for Proteome Biophysics and Chemistry Institute for Functional Materials, Pusan National University, Busan 46241, Korea; suhkmann@pusan.ac.kr; 6Department of Molecular Cell Biology, School of Medicine, Sungkyunkwan University, Suwon 16419, Korea; freefall@skku.edu

**Keywords:** macrophage, cancer, low temperature, glutamine, glutamine synthetase

## Abstract

Ambient temperature can regulate the immune response and affect tumor growth. Although thermoneutral caging reduces tumor growth via immune activation, little attention has been paid to the tumorigenic effect of low temperature. In the present study, tumor growth was higher at low ambient temperature (4 °C for 8 h/d) than at the standard housing temperature (22 °C) in allograft models. Low temperature-stimulated tumor growth in mice was reduced by monocyte depletion using clodronate liposomes. Proliferation was considerably greater in cancer cells treated with 33 °C-cultured RAW264.7 cell-conditioned media (33CM) than in cells treated with 37 °C-cultured RAW264.7 cell-conditioned media (37CM). Additionally, glutamine levels were markedly higher in 33CM-treated cells than in 37CM-treated cells. We further confirmed that the addition of glutamine into 37CM enhanced its effects on cancer cell proliferation and glutamine uptake inhibition ameliorated the accelerated proliferation induced by 33CM. Consistently, the inhibition of glutamine uptake in the allograft model exposed to low temperature, effectively reduced tumor volume and weight. Collectively, these data suggest that the secretion and utilization of glutamine by macrophages and cancer cells, respectively, are key regulators of low temperature-enhanced cancer progression in the tumor microenvironment.

## 1. Introduction

Ambient physical factors, such as temperature and humidity, induce biological events from the cellular to the organismal level [1]. In particular, ambient temperature, which changes depending on longitude, latitude, altitude, and even day and night, can contribute to the risk of cancer mortality [2]. Recent studies have shown that a low-temperature environment may be a major risk factor for several types of cancer, such as lung, bladder, ovarian, skin, stomach, breast, thyroid, and prostate cancer, by increasing their incidence and mortality rates [3,4,5,6]. However, there is little known about the mechanisms responsible for the effects of low ambient temperature on human diseases, especially malignant tumors.

The formation, growth, and metastasis of tumors are markedly decreased in immunocompetent mice housed at thermoneutral temperatures (30–31 °C) compared with those in standard housing temperatures (22–23 °C). These effects occur via an increase in the number of Cd8^+^ T cells and a decrease in the number of myeloid-derived suppressor cells and regulatory T cells [7]. In addition, the co-injection of Lewis lung carcinoma (LLC) cells with adipocytes activated by hypothermic stress promotes allograft cancer cell growth in athymic mice [8]. These studies demonstrated that cells in the tumor microenvironment are important for the accelerated tumor growth induced by low ambient temperature. Although most immune cells infiltrating the tumor site are macrophages [9], there is little known about the role of macrophages in the promotion/progression of tumors induced by low temperature. Previous studies on low temperature-activated macrophages have focused on the beneficial anti-inflammatory effects due to the production of anti-inflammatory cytokines, including interleukin (IL)-10 [10] and on the metabolic refinement of adipose tissue by the secretion of IL-4 and catecholamines [11]. However, IL-4 and/or IL-10 alternatively activate macrophages and polarize them into tumor-associated macrophages, resulting in tumor promotion/progression.

Glutamine is a nutrient that participates in energy metabolism and signal transduction in cancer cells [12]. Cancer and immune cells import glutamine via solute-linked carrier family 1 member A5 (SLC1A5). It then enters the metabolic network to promote cancer cell proliferation and regulate cancer-associated immune cell function [13]. Cancer-associated fibroblasts (CAFs) are one of the most glutamine-producing stromal cells in tumor microenvironment [14]. In addition to fibroblasts, macrophages are considered to be another source of glutamine, via their activation through the classical pathway in the tumor microenvironment [15]. However, little is known regarding glutamine production by macrophages or the roles of macrophages in tumor growth at low ambient temperature.

In this study, we aimed to determine the effects of low ambient temperature on tumor growth using an allograft model. Throughout the in vitro and in vivo experiments, we found that the exposure of macrophages to low temperature promoted tumor growth via increased glutamine secretion. Moreover, the inhibition of glutamine uptake by cancer cells ameliorated low temperature-induced proliferation. Therefore, our findings suggest that enhanced glutamine secretion by macrophages may play a pivotal role in tumor growth induced by low temperature.

## 2. Experimental Section

### 2.1. In Vivo Exposure to Low Ambient Cold Temperature

Male C57BL/6 mice (6 weeks old, 20–24 g) were purchased from Orient Bio Inc. (Seongnam, Korea). Mice were granted free access to a standard diet with drinking water before the experiment. All mice were housed in laboratory cage rack systems maintained at a constant temperature (22 ± 1 °C or 4 ± 1 °C) and humidity (50 ± 5%). The rooms were maintained under a 12 h dark/light cycle. The low ambient temperature condition was induced by 8 h of exposure to 4 °C, daily for 15 d, according to previous reports, with some modifications [11,16]. The mice (n = 8 for each group) were sacrificed by CO_2_ inhalation immediately after the final period of exposure to low temperature at 15 d. Tumor tissues were isolated from all mice for further analyses. Tumor size was measured using a pair of calipers and the volume was calculated using the following formula: (length × width^2^)/2. All experimental procedures followed the Guidelines for the Care and Use of Laboratory Animals of the National Institutes of Health of Korea and were approved by the Institutional Animal Care and Use Committee of Pusan National University (protocol PNU-2019-2184, approved on 13 March 2019).

### 2.2. In Vivo Macrophage Depletion

Mouse macrophages were depleted via intraperitoneal injection of clodronate liposomes (7 mg or 3.5 mg of clodronate per 1 mL; FormuMax Scientific, Sunnyvale, CA, USA) dissolved in phosphate-buffered saline (PBS) at an initial dose of 200 or 100 μL. Three injections were administered, with one injection every 4 d. The efficiency of peripheral blood monocyte depletion was verified using a hematology analyzer.

### 2.3. Cell Culture

LLC cells (CRL-1642) and pre-adipocyte 3T3-L1 cells (CL-173) were obtained from the American Type Culture Collection (Manassas, VA, USA). RAW264.7 cells (KCLB40071), B16F10 melanoma cells (KCLB80008), CT26 colon cancer cells (KCLB80009), and NIH/3T3 fibroblast cells (KCLB21658) were obtained from the Korean Cell Line Bank (KCLB, Seoul, Korea). All cells except CT26 and 3T3-L1 cells were maintained in Dulbecco’s modified Eagle’s medium (DMEM; Gibco, Carlsbad, CA, USA) supplemented with 10% (*v*/*v*) heat-inactivated fetal bovine serum (FBS; Sigma-Aldrich, St. Louis, MO, USA) and antibiotics (100 U/mL penicillin and 100 μg/mL streptomycin; Thermo Fisher Scientific, Rockford, IL, USA). CT26 cells were cultured in RPMI 1640 medium (Gibco) supplemented with 10% (*v*/*v*) heat-inactivated FBS and antibiotics. 3T3-L1 cells were cultured in DMEM supplemented with 10% (*v*/*v*) calf serum (Thermo Fisher Scientific) and antibiotics. The cells were incubated in a humidified incubator at 37 °C under a 5% CO_2_ atmosphere before the experiments.

### 2.4. Metabolite Measurements by CE-TOFMS

Conditioned media (CM) were centrifugally filtered through a 3-kDa cut-off filter (Millipore, Burlington, MA, USA) at 2800× *g* for 120 min at 4 °C for protein removal. The filtrate was prepared using a Milli-Q water system containing an internal standard solution (ULTRAFREE-MC-PLHCC, Human Metabolome Technologies; HMT, Inc., Yamagata, Japan). The samples were analyzed using a TOFMS system (Agilent Technologies Inc., Santa Clara, CA, USA), as described previously [17]. With this system, the two modes of measurement were used to detect both cationic and anionic metabolites. Peak information, including *m/z*, migration time, and peak area determined by CE-TOFMS analysis, was extracted using automatic integration software (MasterHands 2.16.0.15, Keio University, Tsuruoka, Japan). The peaks were assigned using putative metabolites from the HMT metabolite database, according to their MTs in CE and the *m/z* values determined by TOFMS. The tolerance range for peak annotation was configured at ±0.5 min for MT and ±10 ppm for *m/z* (mass error [ppm] = [measured value − theoretical value]/measured value × 10^6^). The peak areas were then converted to relative peak areas according to the equation: metabolite peak area/internal standard peak area.

### 2.5. Statistical Analysis

The data are presented as the mean ± standard deviation (SD) and were analyzed using GraphPad Prism (GraphPad Software Inc., La Jolla, CA, USA). A two-tailed Student’s *t*-test was used for comparisons between two different groups and a one-way or two-way ANOVA, followed by Tukey’s post hoc test, was used for comparisons between multiple groups, as indicated. Differences with a *p*-value < 0.05 were considered significant. All experiments except those in the animal studies were performed on at least three independent occasions.

## 3. Results

### 3.1. Low Ambient Temperature Accelerated Tumor Growth in an Allograft Model

Previous investigations regarding the effects of low ambient temperature on living organisms have focused on thermogenesis and fat browning, but not on cancer biology. To investigate the effect of a cold environment on tumor growth, we exposed tumor-bearing mice to low ambient temperature (4 °C) for 8 h/d (Figure 1A). Among several established models of exposure to low temperatures [11,18,19], we adopted repeated exposure to 4 °C [19], as this experimental condition is considered to be most similar to the climate during winter in cold regions. We found that tumors originating from LLC cells isolated from mice exposed to low temperatures were significantly larger than those originating from LLC cells isolated from control mice (Figure 1B–D). To determine whether the enhanced tumor growth was an LLC cell-specific phenomenon, we performed the same experiments with B16F10 melanoma cells. Interestingly, tumors originating from B16F10 cells grew faster when their host mice had been exposed to low temperature (Figure 1E–G and Appendix A). These findings suggest that low ambient temperature can promote tumor growth.

### 3.2. Depletion of Macrophages Reduced the Growth of Tumors in the Allograft Model

The growth of cancer cells is supported by stromal cells, such as macrophages, fibroblasts, and adipocytes, in the tumor microenvironment [20]. In this study, we hypothesized that macrophages were the major cell type in situ, as their activation by low ambient temperature has previously been reported [11], and they are one of the most common cell types in the tumor microenvironment [9]. To determine the role of macrophages in low temperature-induced tumor growth, we depleted monocyte-lineage cells using clodronate liposomes (Figure 2A). As expected, macrophage depletion markedly reduced the growth of allograft tumors derived from low temperature-exposed LLC cells, together with reduced Cd68^+^ monocyte/macrophage infiltration (Figure 2B–F). These findings suggest that macrophages are a prerequisite for the low temperature-induced acceleration of tumor growth.

### 3.3. Low Temperature-Activated Macrophages Enhanced Cancer Cell Growth

Generally, mammalian cells grow optimally at 37 °C, and low temperatures, such as 33 °C used in our experiments, inhibit their growth [21]. As the surface temperature of the head and extremities is approximately 28–34 °C when exposed to low ambient temperatures [22], and there was decreased in LLC cell growth at 33 °C compared at 37 °C for 2 d (Figure 3A), it was evident that low temperatures did not directly increase the growth of cancer cells. We used 33 °C as the temperature to study the effects of stromal cells on tumor cell growth in subsequent experiments (Figure 3B). LLC cells treated with 33 °C-cultured RAW264.7 cell-conditioned media (33CM) proliferated at a higher rate than those treated with 37 °C-cultured RAW264.7 cell-conditioned media (37CM, Figure 3C). Moreover, conditioned media from bone marrow-derived macrophages (BMDMs) showed similar results to conditioned media from RAW264.7 cells (Figure 3D). The coculture of 33 °C-primed RAW264.7 cells also enhanced the growth of LLC cells (Figure 3E,F). Moreover, the proliferation of B16F10 and CT26 cells was also increased by treatment with 33 cm (Appendix A). Other types of cells, such as fibroblasts and adipocytes, in the tumor microenvironment showed weaker effects on LLC cell growth than macrophages, even though their trends towards enhancing cell growth were similar (Appendix A). Thus, we used the RAW264.7 cells for further studies. Additionally, in the case of hydroxyethyl piperazine ethane sulfonic acid (HEPES)-titrated media to pH 7.4 from RAW264.7 cells cultured at 33 °C or 37 °C, the growth of LLC cells was not changed (Appendix A). These results suggest that the low temperature-stimulated growth of tumors may be linked to activated stromal cells, especially macrophages.

### 3.4. Glutamine Levels Were Elevated in the Media of Macrophages Exposed to Low Temperature

The effect of 33CM on the growth of LLC cells was unchanged after boiling the 33CM (Figure 4A). Furthermore, the proliferation of LLC cells treated with 33CM fractionated using a 3-kDa cut-off filter was similar to that of LLC cells treated with non-fractionated 33CM (Figure 4B). Based on these findings, we speculated that small molecules, rather than macromolecules such as proteins and lipids, may be responsible for the enhanced tumor growth induced by macrophages exposed to low temperature. To determine the critical molecule(s), we performed metabolomic analyses using capillary electrophoresis-time-of-flight mass spectrometry (CE-TOFMS). Metabolomic analyses identified 44 metabolites with increased levels after 33CM treatment. Among them, glutamine showed the greatest upregulation (47-fold) in 33CM-treated cells compared with 37CM-treated cells (Figure 4C and Appendix A). High performance liquid chromatography (HPLC) quantification also showed that glutamine levels were considerably higher in 33CM-treated cells (135.76 μg/mL) than in 37CM-treated cells (5.43 μg/mL, Figure 4D). Additionally, glutamine levels were higher in lysates from RAW264.7 cells cultured at 33 °C (42.80 μg/mL) than those cultured at 37 °C (0.40 μg/mL, Figure 4E). Secreted glutamine levels were lower in both cell lysates and CM derived from NIH/3T3 and differentiated 3T3-L1 cells (Appendix A). These data suggested that macrophages exposed to low ambient temperature mediated tumor cell growth via enhanced glutamine secretion.

### 3.5. Low Ambient Temperature Stimulated Glutamine Secretion in Macrophages by Increasing Glutamine Synthetase (Glul) Protein Levels

The mRNA expression of *Glul* was unaffected by the culture temperature in RAW264.7 cells in contrast to glutaminase *(Gls)* mRNA levels, which decreased with exposure to low temperature (Figure 5A). However, the protein levels of Glul significantly increased in a time-dependent manner in RAW264.7 cells cultured at 33 °C (Figure 5B). Glutamine levels are regulated by the balance between the anabolic enzyme, Glul, and the catabolic enzyme, Gls, and glutamine plays a key role in cancer cell metabolism [12]. As *Gls* mRNA was downregulated in RAW264.7 cells exposed to 33 °C, in contrast to the observed increase in glutamine levels, we silenced *Glul* in RAW264.7 cells using siRNA (Figure 5C,D). Following *Glul* silencing, extracellular and intracellular glutamine levels were significantly reduced in RAW264.7 cells exposed to low temperature (Figure 5E,F).

### 3.6. Extracellular Glutamine Derived from Low Temperature-Exposed Macrophages Enhanced Cancer Cell Growth

The deprivation of glutamine from cell culture media induced cell death rather than inhibiting the proliferation of LLC cells (Appendix A). HPLC analyses showed a difference in glutamine concentration of approximately 1 mM. Therefore, we examined whether an additional 1 mM glutamine could compensate for the decreased growth of cells treated with 37CM. The addition of 37CM containing 1 mM glutamine resulted in an increase in LLC cell growth to the same level as 33CM-treated cells, but glutamine addition had no effect on the growth of 33CM-treated cells (Figure 6A). However, treatment of glutamine uptake inhibitor, l-γ-Glutamyl-p-nitroanilide (GPNA) at nontoxic dose of 1 mM (Appendix A), successfully reduced 33CM-treated LLC cell growth to the level of 37CM-treated cell growth. The expression of glutamine transporter for extracellular secretion, Slc7a5 and Slc7a8, was not significantly changed by culture temperature (Appendix A). Cells treated with 37CM showed similar growth and *Slc1a5* mRNA expression levels, regardless of 1 mM GPNA treatment (Figure 6B and Appendix A). Consistent with these findings, the number of BrdU-positive LLC cells increased after treatment with glutamine-supplemented 37CM and decreased after treatment with GPNA-supplemented 33CM (Figure 6C). The effects of glutamine and GPNA on cell cycle progression were further analyzed by propidium iodide labeling (Figure 6D). Moreover, the expression levels of cell cycle regulatory proteins, including cyclin A2, D1, and E, were consistent with flow cytometric data (Figure 6E).

### 3.7. Inhibition of Glutamine Uptake Attenuated Low Temperature-Accelerated Allograft Tumor Growth

The results of the present study indicate that low ambient temperature promotes glutamine secretion from macrophages in the tumor microenvironment, which accelerates cancer cell growth. Based on these findings, we examined whether the inhibition of glutamine uptake could ameliorate the acceleration of tumor growth induced by low temperature (Figure 7A). GPNA successfully inhibited LLC allograft tumor growth at low temperatures, compared to tumor growth in the control group (Figure 7B–D and Appendix A), without affecting body weight (Appendix A). Concomitantly, serum glutamine levels were higher in the low ambient temperature group (73.85 μg/mL) than the control group (43.95 μg/mL), regardless of GPNA administration (Figure 7E). Moreover, the expression levels of cell cycle regulatory proteins were consistent with those observed in the in vitro experiments (Figure 7F). Collectively, our findings indicate that increased glutamine levels in the tumor microenvironment, mainly produced by macrophages, play a pivotal role in tumor growth induced by low ambient temperature.

## 4. Discussion

In the present study, we showed that low ambient temperature accelerated tumor cell growth in vitro and in vivo. Among the several types of stromal cells in the tumor microenvironment, we identified macrophages as prerequisites for low temperature-accelerated tumor growth. We further found that small molecules, rather than macromolecules, were important in this phenomenon. Metabolite analyses and pharmacological in vivo experiments showed that glutamine was a pivotal molecule among the small molecules identified to be potentially involved in low temperature-enhanced tumor growth.

Several tumor stromal cell types, such as fibroblasts and adipocytes, are known to support tumor cell growth at low temperatures [8]. However, less is known about the role of macrophages in low temperature-accelerated tumor growth, even though macrophages and/or their precursor cells, monocytes, may be directly exposed to low ambient temperatures. Exposure to low temperatures stimulates stromal cells in the peripheral blood and skin to increase their metabolism [23]. In this study, we adopted a co-culture system and monocyte/macrophages conditioned media transfer to mimic the role(s) of these cells in human abdominal solid tumors and/or to overcome our subcutaneous allograft tumor model. To confirm that macrophages play a pivotal role in low temperature-accelerated tumor growth, we evaluated not only LLC cells, but also B16F10 and CT26 cells, which represent melanoma and colorectal cancer models, respectively. Taken together, the data from the present study indicate that the impact of low ambient temperature on tumor growth was largely directly associated with the communication between cancer cells and tumor-associated macrophages.

Following the identification of macrophages as the major cell type involved in the effects of low ambient temperature, we speculated that certain small metabolites were the key molecules responsible for enhancing tumor growth in 33CM-treated cells. Metabolomic and quantitative HPLC analyses of soluble factors secreted by macrophages exposed to low temperature showed that glutamine levels were markedly increased. Glutamine is generally regarded as a potential target for cancer therapy to slow cancer cell growth [24]. Blocking de novo glutamine synthesis by suppressing *Glul* expression in CAFs reverses the growth of glutamine-dependent cancer cells [14]. Our results also demonstrate that intra- and extracellular glutamine levels were decreased by the knockdown of *Glul* in 33CM-treated RAW264.7 cells. These results suggest that targeting glutamine synthesis may be an effective strategy in macrophages as well as CAFs.

In a low-temperature environment, Glul levels may not be simply regulated at the transcriptional level. Previous studies have reported increased Glul activity after exposure of animals including rats, hedgehogs, chicks, and rainbow smelt to low temperatures [25,26,27,28]. In plants, low temperatures increase the protein levels of Glul [29,30]. The protein stability of Glul could be controlled by the ubiquitin-dependent proteasomal degradation mechanism [31]. It has known that p300/CBP acetylation of Glul binds cereblon, resulting in ubiquitylation by CRL4 and degradation of glutamine [32]. In addition, glutamine or γ-aminobutyric type B receptors can regulate the protein level of Glul [33,34]. In *Medicago truncatula*, phosphorylation of Glul catalyzed by a calcium-dependent protein kinase and 14-3-3 interaction also regulates its protein stability [35]. Therefore, we assume that increased stability of Glul protein in low temperature-activated macrophages will influence the glutamine secretion. However, because of the complexity of the regulatory mechanism(s) of Glul protein stability, more extensive studies are required to clarify the precise mechanism underlying their regulation by low temperature.

Despite the physiological importance of glutamine [36], low temperature-induced glutamine production has received little attention in the field of cancer metabolism. Cancer cells require glutamine to meet their metabolic requirements to support rapid proliferation and energy production [37]. SLC1A5 acts as a high-affinity transporter of l-glutamine in rapidly growing cancer cells [38]. Blocking SLC1A5 to inhibit glutamine uptake successfully suppresses lung tumor cell proliferation [39]. Glutamine is also a precursor of nucleotide biosynthesis and mitochondrial bioenergetics. In the present study, GPNA treatment sufficiently blocked low temperature-induced tumor cell growth in vitro and in vivo. As demonstrated in previous studies [25,26,27,28], we found an increase in serum glutamine levels after exposure to low temperature, regardless of GPNA administration. We suspect that the ornithine cycle in the liver and increased urination to maintain blood osmolarity may have contributed to the effect(s) of GPNA on serum glutamine levels [40].

## 5. Conclusions

We found that low ambient temperature stimulated glutamine secretion in tumor-associated macrophages to support tumor cell growth. Inhibiting glutamine synthesis in macrophages or blocking glutamine utilization by cancer cells reduced low temperature-enhanced cancer growth. Moreover, protection of cancer patients from low ambient temperature may slow cancer progression. Based on our findings, a greater understanding of the interaction between ambient temperature and cancer metabolism may assist in the development of strategies to decrease cancer mortality rates.

## Figures and Tables

**Figure 1 biomedicines-08-00381-f001:**
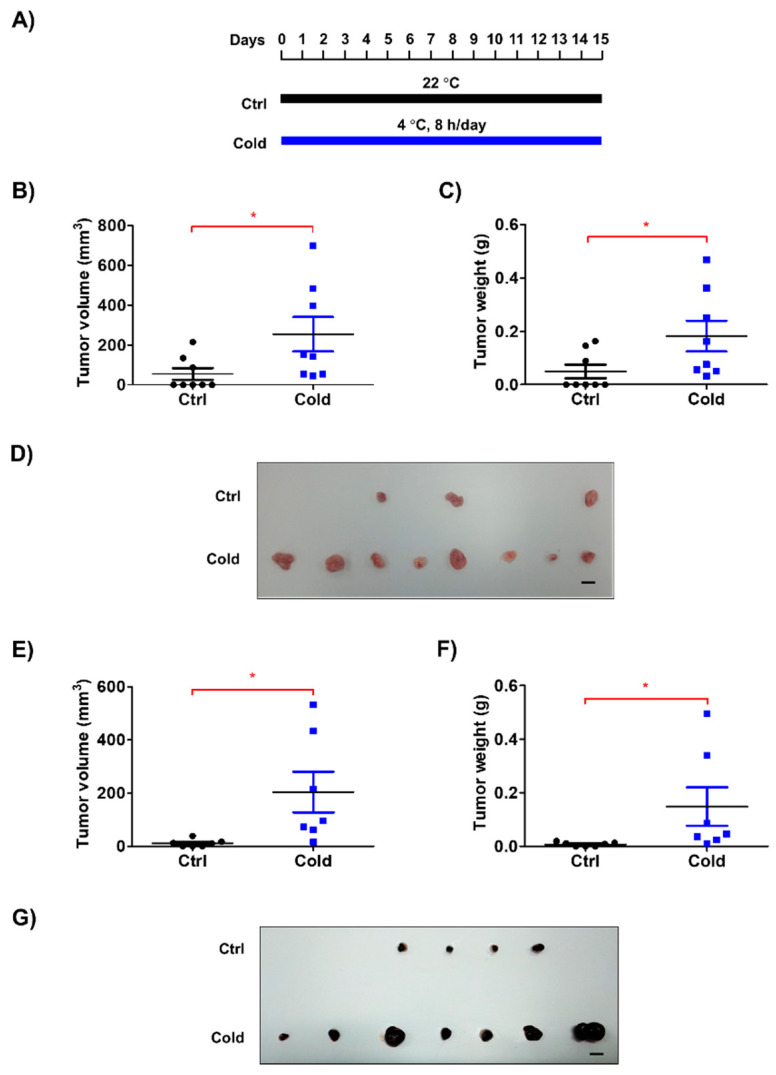
Low ambient temperature accelerated the growth of tumors in an allograft model. (**A**) Lewis lung carcinoma (LLC) cells (1 × 10^5^ cells/100 μL PBS) and B16F10 cells (5 × 10^4^ cells/100 μL PBS) were subcutaneously injected into the dorsum of mice. The mice were housed at 22 °C and mice in the cold group were exposed to 4 °C, 8 h/d for 15 d. (**B**,**C**,**E**,**F**) After 15 d, tumor volume and weight were measured. (**D**,**G**) Images of tumor samples from the control and cold groups are shown. The bar indicates 1 cm. Data are expressed as the mean ± SD. * *p* < 0.05 compared to the control group (22 °C). SD, standard deviation; Ctrl, control.

**Figure 2 biomedicines-08-00381-f002:**
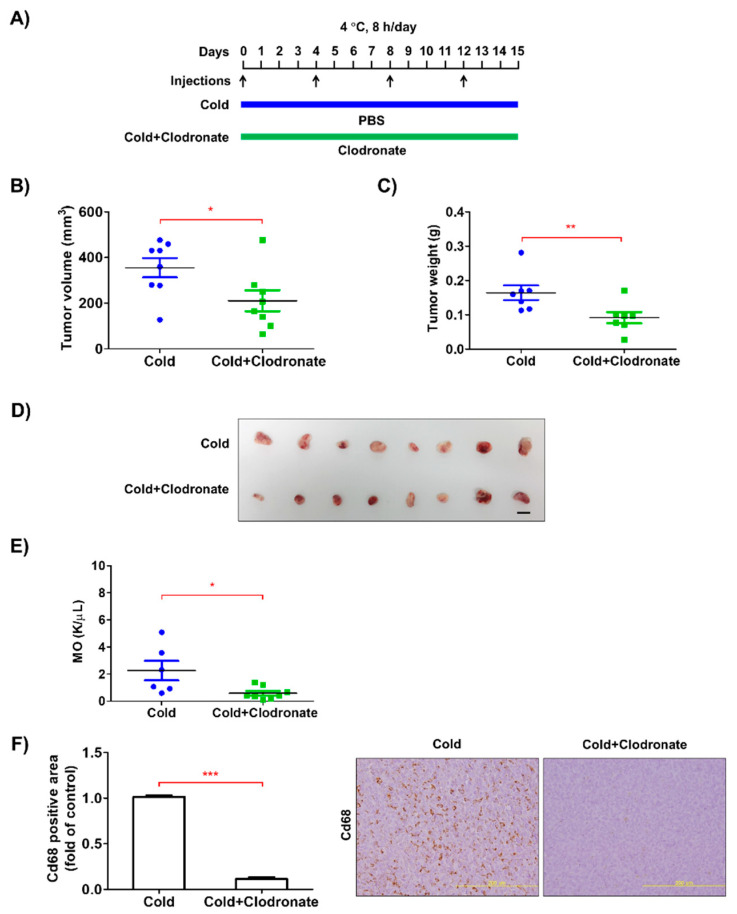
Macrophage depletion reduced low temperature-induced growth of allograft LLC cells. (**A**) LLC cells (1 × 10^5^ cells/100 μL PBS) were subcutaneously injected into the dorsum of mice. Clodronate liposomes were intraperitoneally injected into the mice at 7 mg/kg (first dose), followed by 3.5 mg/kg every 4 d for 15 d. PBS was injected as a control. (**B**,**C**) After 15 d, tumor volume and weight were measured. (**D**) Images of tumor samples from each group are presented. The bar indicates 1 cm. (**E**) Monocytes in peripheral blood were counted using a hematology analyzer. (**F**) The Cd68-positive area of tumor tissues was calculated by immunohistochemical analysis. Representative microscopic images (× 200) are shown. Data are expressed as the mean ± SD. * *p* < 0.05, ** *p* < 0.01 and *** *p* < 0.001 compared to the control group (PBS).

**Figure 3 biomedicines-08-00381-f003:**
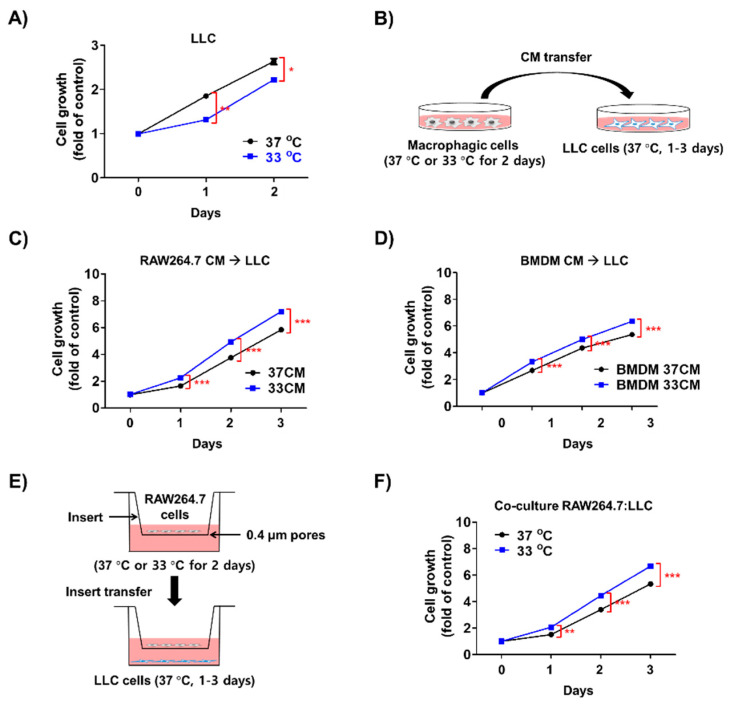
The proliferation of LLC cells was enhanced by macrophages exposed to low temperature. (**A**) LLC cells were incubated at 33 or 37 °C for 48 h. Cell growth was measured by MTT assay. (**B**) A schematic representation of serum-transfer experiments. Culture media from RAW264.7 cells or BMDMs incubated at 33 or 37 °C for 48 h were transferred to LLC cells, which were then cultured 37 °C for 72 h. (**C**,**D**) The growth of LLC cells in the presence of conditioned media (CM) from RAW264.7 cells (**C**) or bone marrow-derived macrophages (BMDMs) (**D**) was measured by MTT assay for 72 h. (**E**) Schematic presentation of the co-culture of RAW264.7 cells and LLC cells. Co-culture was performed by seeding RAW264.7 cells in the upper chamber and plating LLC cells in the lower chamber. (**F**) The growth of LLC cells co-cultured with RAW264.7 cells was examined by MTT assay. Data are expressed as the mean ± SD. * *p* < 0.05, ** *p* < 0.01, and *** *p* < 0.001 compared to the control (37 °C).

**Figure 4 biomedicines-08-00381-f004:**
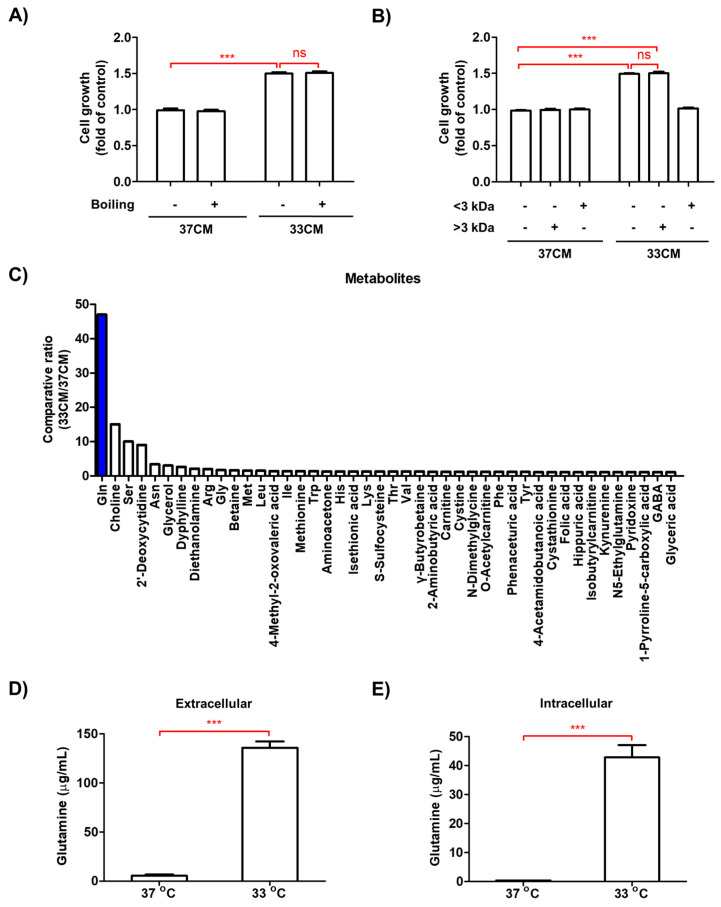
Glutamine production was elevated in macrophages exposed to low temperature. (**A**) CM from RAW264.7 cells was boiled for 10 min. The growth of LLC cells in the presence of RAW264.7 cell CM that was either boiled or untreated was measured by MTT assay. (**B**) CM from RAW264.7 cells was fractionated using a 3 kDa cut-off filter. LLC cells were incubated with RAW264.7 CM size fractionated at <3 kDa or >3 kDa, or with whole CM. The growth of LLC cells was examined using an MTT assay. (**C**) Metabolite levels in RAW264.7 CM were measured by CE-TOFMS. The ratio of metabolites in 33CM/37CM is presented as the average value. (**D**,**E**) Glutamine levels in CM (**D**) and cell lysates (**E**) of RAW264.7 cells were measured using HPLC analysis. Data are expressed as the mean ± SD. *** *p* < 0.001 compared to the control (37 °C). ns means no significance.

**Figure 5 biomedicines-08-00381-f005:**
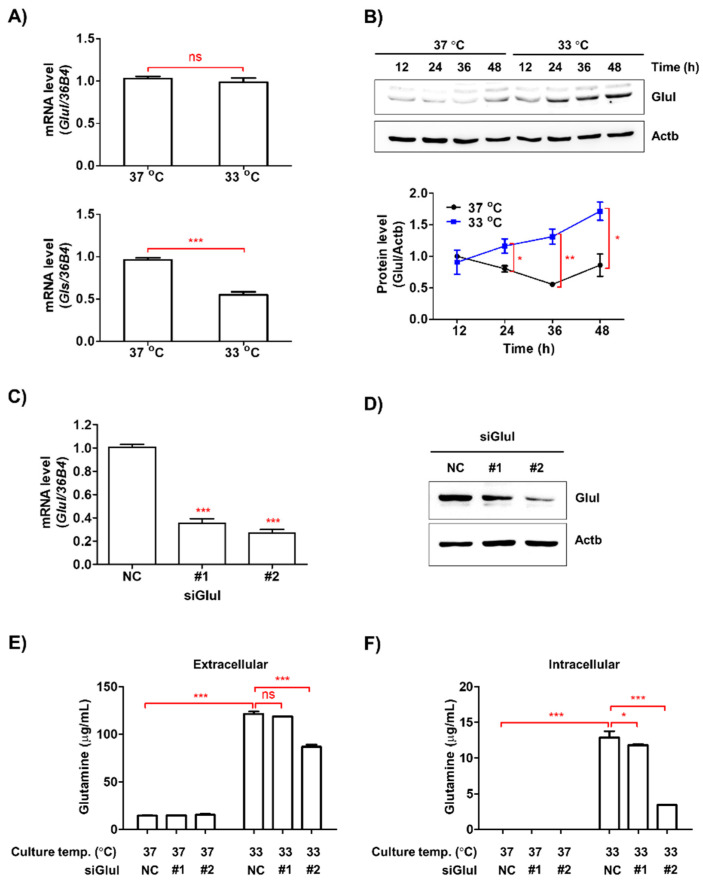
Knockdown of *Glul* decreased glutamine levels in macrophages exposed to low temperature. (**A**) RAW264.7 cells were incubated at 33 °C or 37 °C for 48 h. The mRNA expression levels of *Glul* and *Gls* were determined by qRT-PCR. *36B4* was used as an internal control. (**B**) The protein levels of Glul in RAW264.7 cells were measured by Western blotting analysis. Actin was used as a loading control. (**C**,**D**) Glul expression levels in the RAW264.7 cells transfected with scramble siRNA (NC) or two specific siRNA (#1 and #2) for *Glul* were determined by qRT-PCR and Western blotting analyses. (**E**,**F**) Glutamine levels in CM (**E**) and cell lysates (**F**) of *Glul* siRNA-transfected RAW264.7 cells were measured using HPLC analysis. Data are expressed as the mean ± SD. * *p* < 0.05, ** *p* < 0.01, and *** *p* < 0.001 compared to the control (37 °C). ns means no significance.

**Figure 6 biomedicines-08-00381-f006:**
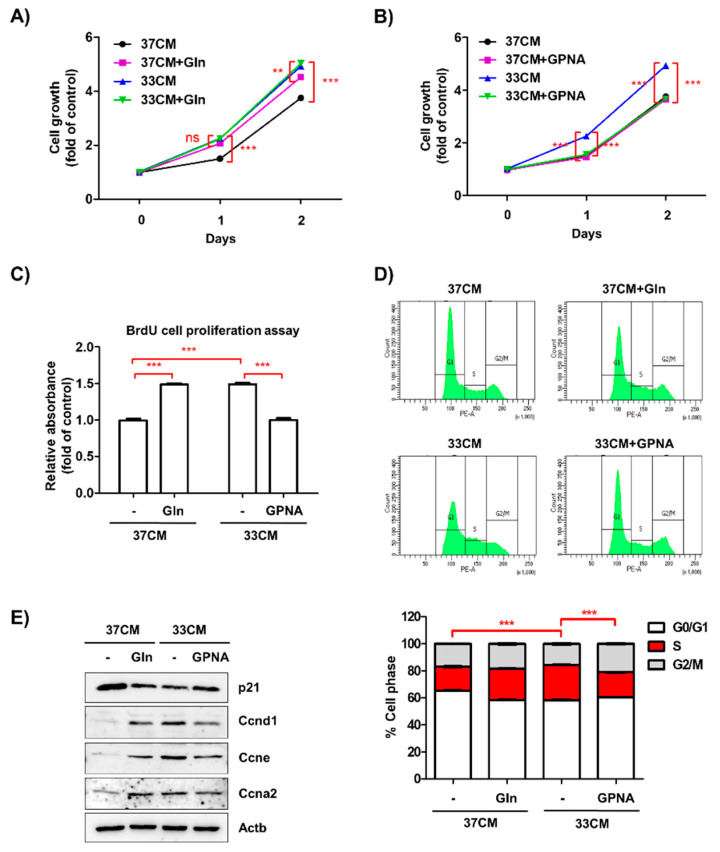
l-γ-Glutamyl-p-nitroanilide (GPNA) suppressed the growth of cancer cells induced by macrophages exposed to low temperature. (**A**,**B**) LLC cells were incubated with RAW264.7 CM every day for 48 h, with the addition of 1 mM glutamine (+Gln) or GPNA (+GPNA), for panels A and B, respectively. The growth of LLC cells was examined using an MTT assay. (**C**) The effects of Gln or GPNA treatment on the proliferation of LLC cells in the presence of CM from RAW264.7 cells were analyzed using a BrdU incorporation assay. (**D**) LLC cells (5 × 10^4^ cells/58 cm^2^) were incubated with RAW264.7 CM for 24 h. The cell cycle stage was determined by fluorescent-activated cell sorting (FACS) analysis using propidium iodine (PI) staining. Representative histograms of the cell cycle distribution of LLC cells are shown. (**E**) Western blotting analysis results of p21, cyclin D1, cyclin E, and cyclin A2 expression in LLC cells. Actin was used as a loading control. The results are expressed as the mean ± SD. ** *p* < 0.01 and *** *p* < 0.001 compared between two groups. ns means no significance.

**Figure 7 biomedicines-08-00381-f007:**
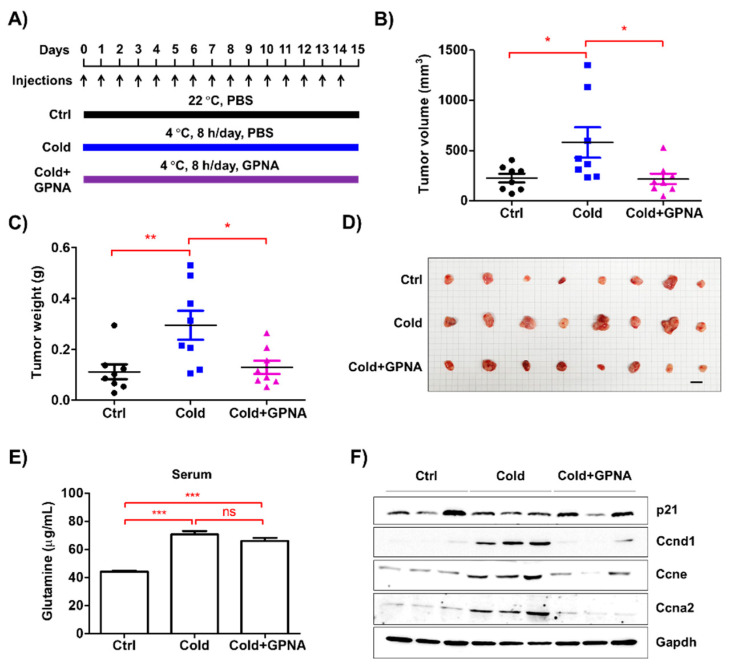
GPNA inhibited allograft tumor growth in low ambient temperature conditions. (**A**) LLC cells (1 × 10^6^ cells/100 μL PBS) were subcutaneously injected into the dorsum of mice. GPNA (20 mg/kg) was intraperitoneally injected into the mice every day for 15 d. (**B**,**C**) After 15 d, tumor volume and weight were measured. (**D**) Images of tumor samples from the ctrl, cold, and cold+GPNA groups are shown. The bar indicates 1 cm. (**E**) Serum glutamine levels were measured using HPLC analysis. (**F**) Western blotting analysis results of p21, cyclin D1, cyclin E, and cyclin A2 expression in tumor tissues from three representative mice in each group. GAPDH was used as a loading control. Data are expressed as the mean ± SD. * *p* < 0.05, ** *p* < 0.01, and *** *p*< 0.001 compared to the control group (22 °C). ns means no significance.

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
