# Peer review of "Macrophage Stimulated by Low Ambient Temperature Hasten Tumor Growth via Glutamine Production"

_biomedicines, 2020, doi:10.3390/biomedicines8100381_

Round 1

Reviewer 1 Report

The manuscript entitled "Macrophage stimulated by low ambient temperature hasten tumor growth via glutamine production" is comprehensive, with a lot of interesting results. 

Minor revisions:

  1. To improve clarity, it would be useful to explain the abbreviations when first used in the text.
  2. Page 5, line 22; page 6, line  1- there is a discrepancy: "there was no significant decrease in LLC cell growth at 33°C (Figure 3a)", but the changes on figure 3a are significant on days 1 (++) and 2 (+). 
  3. Page 6, line 7: figure 3d, not 2d
  4. Page 6, line 8: figures 3ef, not 2ef
  5. Please unify the way in which references are cited. Use the standard journal abbreviations. 

Author Response

Answers to Reviewer #1:

Comments and Suggestions for Authors

The manuscript entitled "Macrophage stimulated by low ambient temperature hasten tumor growth via glutamine production" is comprehensive, with a lot of interesting results.

Minor revisions:

Question 1. To improve clarity, it would be useful to explain the abbreviations when first used in the text.

Answer: As indicated by Reviewer, we added the abbreviations when first used in the text.

Question 2. Page 5, line 22; page 6, line 1- there is a discrepancy: "there was no significant decrease in LLC cell growth at 33°C (Figure 3a)", but the changes on figure 3a are significant on days 1 (++) and 2 (+).

Page 6, line 7: figure 3d, not 2d

Page 6, line 8: figures 3ef, not 2ef

Answer: Thank you very much for Reviewer’s kind comments. As pointed by Reviewer, the paragraph of “Results” section was modified, as follow:

Page 5, line 22; and there was decreased in LLC cell growth at 33°C compared at 37°C for 2 d (Figure 3a), it was evident that low temperatures did not directly increase the growth of cancer cells.

Page 6, line 7: Moreover, conditioned media from bone marrow-derived macrophages (BMDMs) showed similar results to conditioned media from RAW264.7 cells (Figure 3d).

Page 6, line 8: The coculture of 33 C-primed RAW264.7 cells also enhanced the growth of LLC cells (Figure 3e,f).

Beside these, other typographical errors were also corrected and indicated by red characters.

Question 3. Please unify the way in which references are cited. Use the standard journal abbreviations.

Answer: As pointed out by Reviewer, we overall revised the references and standard journal abbreviations in manuscript and supplementary and indicated by red characters.

Reviewer 2 Report

The authors investigated the effects of low ambient temperature on tumor growth using an allograft model and macrophages. Tumor growth at 4°C was higher than at 22°C in allograft model. Cell proliferation in LLC cells was faster when treated with 33°C-cultured RAW264.7 cell-conditioned media than 37°C-cultured RAW264.7 cell-conditioned media. Interestingly, glutamine levels in the medium of 33°C-cultured RAW264.7 cell-conditioned media were higher than 37°C-cultured RAW264.7 cell-conditioned media. Glutamine enhanced cell proliferation and glutamine receptor inhibitor inhibition cleared enhanced cell proliferation. The results are sound. However, to mention the current conclusion, some additional studies are needed. Especially, the change of pH level in the medium when altered the temperature, should be investigated. There are concerns that should be addressed.

  1. Figure number in section 3.3. is incorrect. Please improve.
  2. When changed the temperature, pH may be altered. pH level in the medium was measured? The effect of pH in this study should be examined.
  3. Glutamate receptors were expressed in LLC cells under various temperature conditions? It should be investigated.
  4. Why glutamine synthase (GLUL) protein level, but not mRNA level was enhanced by low temperature condition? It’s critical point. Although some descriptions were found in Discussion part, the possibility should be more discussed. The stability of GLUL protein is increased in the low temperature condition?

Author Response

Answers to Reviewer #2:

Comments and Suggestions for Authors

The authors investigated the effects of low ambient temperature on tumor growth using an allograft model and macrophages. Tumor growth at 4°C was higher than at 22°C in allograft model. Cell proliferation in LLC cells was faster when treated with 33°C-cultured RAW264.7 cell-conditioned media than 37°C-cultured RAW264.7 cell-conditioned media. Interestingly, glutamine levels in the medium of 33°C-cultured RAW264.7 cell-conditioned media were higher than 37°C-cultured RAW264.7 cell-conditioned media. Glutamine enhanced cell proliferation and glutamine receptor inhibitor cleared enhanced cell proliferation. The results are sound. However, to mention the current conclusion, some additional studies are needed. Especially, the change of pH level in the medium when altered the temperature, should be investigated. There are concerns that should be addressed.

Question 1. Figure number in section 3.3. is incorrect. Please improve.

Answer: We are sorry for the mistakes. As pointed by Reviewer, the paragraph of “Results” section was modified, as follow:

Generally, mammalian cells grow optimally at 37 C, and low temperatures, such as 33°C used in our experiments, inhibit their growth [21]. As the surface temperature of the head and extremities is approximately 28–34°C when exposed to low ambient temperatures [22] and there was decreased in LLC cell growth at 33°C compared at 37°C for 2 d (Figure 3a), it was evident that low temperatures did not directly increase the growth of cancer cells. We used 33 C as the temperature to study the effects of stromal cells on tumor cell growth in subsequent experiments (Figure 3b). LLC cells treated with 33°C-cultured RAW264.7 cell-conditioned media (33CM) proliferated at a higher rate than those treated with 37°C-cultured RAW264.7 cell-conditioned media (37CM, Figure 3c). Moreover, conditioned media from bone marrow-derived macrophages (BMDMs) showed similar results to conditioned media from RAW264.7 cells (Figure 3d). The coculture of 33 C-primed RAW264.7 cells also enhanced the growth of LLC cells (Figure 3e,f).

Question 2. When changed the temperature, pH may be altered. pH level in the medium was measured? The effect of pH in this study should be examined.

Answer: Thank you for your deliberated comments. We agree to Reviewer’s opinion on the effect of temperature on pH of culture media. We examined pH level of conditioned media from RAW264.7 cells (1 × 105 cells/58 cm2) cultured at 33°C or 37°C for 2 days, in presence of 5% CO2. The pH of 33°C-cultured media (7.57) was slightly increased compared with that of 37°C (7.27). However, there are no statistical significance.

In addition, to examine the effect of pH on the tumor cell growth, we treated with HEPES-titrated media from RAW264.7 cells at 33°C or 37°C. The growth of LLC cells was not changed. We added the result in Figure S2E and revised the supplementary and manuscript, as follow:

In Figure S2E.

Figure S2. (e) CM from RAW264.7 cells was titrated to pH 7.4 with HEPES buffer.

In Result section

Additionally, in the case of hydroxyethyl piperazine ethane sulfonic acid (HEPES)-titrated media to pH 7.4 from RAW264.7 cells cultured at 33°C- or 37°C, the growth of LLC cells was not changed (Figure S2e).

Question 3. Glutamate receptors were expressed in LLC cells under various temperature conditions? It should be investigated.

Answer: As suggested by Reviewer, glutamate receptor could be changed by temperature. In our study, we focused to glutamine because it was highly changed metabolite in metabolomic analysis. However, as inspired by Reviewer’s comment, the expression of glutamine transporter was also changed in different temperature conditions. The expression of Slc1a5, a glutamine transporter for cellular uptake, was detected in LLC cells and did not showed significant changes by treatment of culture media from RAW264.7 cells cultured at 33°C- or 37°C (Figure S4c). In addition, we confirmed the mRNA expression of glutamine transporters for cellular export, Slc7a5 and Slc7a8, in RAW264.7 cells, because the expression of their transporter might influence on the glutamine secretion. There was no significant change by the culture temperature in RAW264.7 cells. The Figure S4, legend, and related Result section were modified as follow.

Figure and legend:

Figure S4. The effect of glutamine or GPNA on the growth of cancer cells. (a) LLC cells were incubated in complete media (Gln+, corresponding to glutamine 584 μg/mL) or glutamine-free media (Gln-) at 37 C for 48 h. The growth of LLC cells was examined using an MTT assay. (b) LLC cells were treated with the indicated concentration of GPNA for 48 h and the growth of LLC cells were measured by MTT assay. (c, d) The RAW264.7 cells were cultured at 33 C or 37 C and incubated for 48 h. The expressions of Slc7a5 and Slc7a8 were measured by qRT-PCR. The 36B4 was used as internal control. (e) The LLC cells were treated with RAW264.7 CM cultured at 33 C or 37 C and incubated for 24 h. The mRNA expression of Slc1a5 was determined by qRT-PCR. The 36B4 was used as internal control. Data are expressed as the mean ± SD. **p < 0.01 and ***p < 0.001 compared to the control. ns mean no significance.

Result section:

However, treatment of glutamine uptake inhibitor, L-γ-Glutamyl-p-nitroanilide (GPNA) at nontoxic dose of 1 mM (Figure S4b), successfully reduced 33CM-treated LLC cell growth to the level of 37CM-treated cell growth. The expression of glutamine transporter for extracellular secretion, Slc7a5 and Slc7a8, was not significantly changed by culture temperature (Figure S4c,d). Cells treated with 37CM showed similar growth and Slc1a5 mRNA expression levels, regardless of 1 mM GPNA treatment (Figure 6b and S4e).

Question 4. Why glutamine synthase (GLUL) protein level, but not mRNA level was enhanced by low temperature condition? It’s critical point. Although some descriptions were found in Discussion part, the possibility should be more discussed. The stability of GLUL protein is increased in the low temperature condition?

Answer: We totally agree to Reviewer’s opinion on the glutamine synthetase (Glul). As pointed by Reviewer, we modified the “Discussion” section to the protein expression of Glul, as follow:

In a low-temperature environment, Glul levels may not be simply regulated at the transcriptional level. Previous studies have reported increased Glul activity after exposure of animals including rats, hedgehogs, chicks, and rainbow smelt, to low temperatures [25-28]. In plants, low temperatures increase the protein levels of Glul [29,30]. The protein stability of Glul could be controlled by the ubiquitin-dependent proteasomal degradation mechanism [31]. It has known that p300/CBP acetylation of Glul binds cereblon, resulting in ubiquitylation by CRL4 and degradation of glutamine [32]. In addition, glutamine or γ-Aminobutyric Type B Receptors can regulate the protein level of Glul [33,34]. In Medicago truncatula, phosphorylation of Glul catalysed by a calcium-dependent protein kinase and 14-3-3 interaction also regulates its protein stability [35]. Therefore, we assume that increased stability of Glul protein in low temperature-activated macrophages will be influence on the glutamine secretion. However, because of the complexity of the regulatory mechanism(s) of Glul protein stability, more extensive studies are required to clarify the precise mechanism underlying their regulation by low temperature.

Reviewer 3 Report

  • Author are advised to mention the rationale to use various cell line from different origin, used in their study. Also, why RAW264.7 cell line conditioned media was used?
  • The authors are advised to mention methodology used in figure 2e and 2f in detail. As it is not clear whether the MTT assay of LLC cells was done after incubating with RAW264.7 cell or together with RAW264.7 while co-culturing.
  • At number of places the methodology of the experiment is not described well. The authors are advised to mention the method of the experiments in detail, in experimental section of the manuscript.

Author Response

Answers to Reviewer #3:

Comments and Suggestions for Authors

Question 1. Author are advised to mention the rationale to use various cell line from different origin, used in their study. Also, why RAW264.7 cell line conditioned media was used?

Answer: Thank you very much for Reviewer’s kind comments. As pointed out by Reviewer, in our study, we represented that ambient low temperature enhanced the growth of several types of cancer, such as lung, melanoma, and colon in vivo. In addition, we showed that low temperature-cultured media from different types of cells which are consist of tumor stroma, such as macrophage-like RAW264.7 cells, fibroblast NIH/3T3, and differentiated adipocyte 3T3-L1 cells. The treatment of cultured media from all these cells cultured at 33°C increased the growth of cancer cells (Figure S2). However, the effect of 33°C-cultured media from RAW264.7 cells on the growth of tumor cells is much higher than that from other cells. The effect of culture media from bone-marrow derived macrophages is similar to that of cultured media from RAW264.7 cells. Thus, we used the RAW264.7 cells for further studies. The related-“Result” section was modified as follow.

Moreover, the proliferation of B16F10 and CT26 cells was also increased by treatment with 33CM (Figure S2a,b). Other types of cells, such as fibroblasts and adipocytes, in the tumor microenvironment showed weaker effects on LLC cell growth than macrophages, even though their trends towards enhancing cell growth were similar (Figure S2c,d). Thus, we used the RAW264.7 cells for further studies.

Question 2. The authors are advised to mention methodology used in figure 2e and 2f in detail. As it is not clear whether the MTT assay of LLC cells was done after incubating with RAW264.7 cell or together with RAW264.7 while co-culturing.

Answer: As pointed by Reviewer, we presented the methodology of “RAW264.7 and LLC cells co-culture” in supplementary method, as follow:

RAW264.7 and LLC cells co-culture

For co-culture experiments, RAW264.7 cells were cultured on the 0.4 μm pore size transwell insert (Corning Co., Corning, NY) and LLC cells were cultured in the bottom well of the transwell chamber. RAW264.7 cells (1 × 105 cells/4.52 cm2) were cultured at 37 C or 33 C for 48 h. Cancer cells (5 × 103 cells/9.6 cm2) were cultured at 37 C or 33 C for 72 h. Then, RAW264.7 cells in the upper insert were added on the top of each 6-well and cultured every day for 72 h.

Question 3. At number of places the methodology of the experiment is not described well. The authors are advised to mention the method of the experiments in detail, in experimental section of the manuscript.

Answer: As pointed out by Reviewer, we overall revised the manuscript and supplementary method and indicated as red characters.

Round 2

Reviewer 2 Report

The manuscript was improved.